# Enhancing Urban Above-Ground Vegetation Carbon Density Mapping: An Integrated Approach Incorporating De-Shadowing, Spectral Unmixing, and Machine Learning

**Guangping Qie [1,2,3], Jianneng Ye [4], Guangxing Wang [3] and Minzi Wang [5,*]**

1   Department of Tourism Management, Moutai Institute, Renhuai 551801, China; qieguangping@mtxy.edu.cn
2   Department of Geography and Environmental Resources, Southern Illinois University at Carbondale, Carbondale, IL 62901, USA
3   School of Earth Systems and Sustainability, Southern Illinois University at Carbondale, Carbondale, IL 62901, USA; gxwang@siu.edu
4   Department of Student Affairs, Zhejiang Gongshang University, Hangzhou 310018, China; jiannengye@mail.zjgsu.edu.cn
5   Department of Resource and Environment, Moutai Institute, Renhuai 551801, China
*   Correspondence: wangminzi@mtxy.edu.cn; Tel.: +86-1830-0942-108

**Abstract:** Accurately mapping urban above-ground vegetation carbon density presents challenges due to fragmented landscapes, mixed pixels, and shadows induced by buildings and mountains. To address these issues, a novel methodological framework is introduced, utilizing a linear spectral unmixing analysis (LSUA) for shadow removal and vegetation information extraction from mixed pixels. Parametric and nonparametric models, incorporating LSUA-derived vegetation fraction, are compared, including linear stepwise regression, logistic model-based stepwise regression, k-Nearest Neighbors, Decision Trees, and Random Forests. Applied in Shenzhen, China, the framework integrates Landsat 8, Pleiades 1A & 1B, DEM, and field measurements. Among the key findings, the shadow removal algorithm is effective in mountainous areas, while LSUA-enhanced models improve urban vegetation carbon density mapping, albeit with marginal gains. Integrating kNN and RF with LSUA reduces errors, and Decision Trees, especially when integrated with LSUA, outperform other models. This study underscores the potential of the proposed framework, particularly the integration of Decision Trees with LSUA, for advancing the accuracy of urban vegetation carbon density mapping.

**Keywords:** urban; vegetation carbon density; mapping; de-shadow; spectral unmixing; machine learning





## 1. Introduction

Urbanization in China, with 65.2% of the population residing in urban areas [1], has led to rapid economic development and increased migration to cities, resulting in reduced vegetation cover and elevated greenhouse gas emissions [2]. Urban areas have become significant contributors to climate change, emphasizing the urgency of understanding and managing urban vegetation carbon density [3]. Recognizing the crucial role of urban vegetation as a carbon sink, accurate estimates of carbon storage and sequestration are essential for informed decision making by governments. Despite increasing awareness of environmental issues, mapping urban vegetation carbon density faces challenges due to complex landscapes, mixed pixels, and mountain- and building-induced shadows [4].

Carbon storage in urban vegetation, predominantly facilitated by urban forests, is a pivotal element in carbon sequestration [5]. Urban forests, encompassing both woody and associated herbaceous plants within and surrounding settlements, demonstrate the capacity to sequester substantial carbon in both above-ground and below-ground biomass [6]. The estimation of carbon levels at various scales involves a diverse array of methods, with local-scale assessments relying on a combination of field and remote sensing data [7–10]. These

assessments take into account factors such as tree species, canopy structure, diameters at breast height (DBH), spatial resolution, spectral bands, and sensor characteristics, etc. Moving to a regional scale, estimates incorporate additional variables related to climate and the environment [11–14]. Commonly, allometric equations, grounded in physiological relationships, are employed for biomass calculations, and their specifics may vary by region [15–20]. While specific allometric equations for urban forests are limited, alternative methods encompass the utilization of volume tables for tree biomass and the transfer of biomass calculations for shrubs and grass in green urban areas. Remote sensing techniques offer a cost-effective strategy for assessing vegetation biomass or carbon density across extensive areas. The extensive adoption of various satellite data sources such as Landsat Thematic Mapper, Landsat 8, Sentinel-2, MODIS, among others, provides multispectral images and seamless integration with Geographic Information Systems (GIS). Various techniques, including spatial interpolation and regression analyses, synergize field data with remote sensing for mapping forest carbon density [21–24]. Remote sensing-based methods, especially utilizing optical sensors like Landsat, MODIS, SPOT, and ALOS, offer promising avenues for mapping vegetation biomass or carbon density [25–27]. Landsat, with its free availability, medium spatial resolution, and historical data, is widely used, yet may underestimate biomass in dense tropical rainforests [7]. Coarse spatial resolution MODIS images suit global estimates but are unsuitable for small areas [28]. Very fine spatial resolution images, e.g., QuickBird and Worldview series, are suitable for urban mapping but hindered by high costs [29]. Active remote sensing techniques, such as LiDAR and TLS, offer advantages in penetrating clouds but pose challenges in cost and sensitivity to certain forest types [30–33]. Urban vegetation mapping is particularly challenging due to fragmented landscapes and building- and mountain-induced shadows, with limited research in this domain.

Fragmented urban landscapes often exhibit mixed pixels, complicating the effective use of remote sensing data for Land Use and Land Cover (LULC) or change detection analysis [34,35]. Spectral unmixing analysis, also known as Spectral Mixture Analysis or Spectral Mixture Modeling, addresses this challenge by extracting sub-pixel information through decomposing mixed pixels into fractional images corresponding to each endmember, representing a pure LULC type [36–40]. Spectral unmixing analysis methods fall into two categories based on the algorithms used: linear mixture models (LMMs) and nonlinear mixture models (NLMMs). LMMs assume that a mixed pixel's DN value is determined by the weights of endmember spectra and corresponding coverage area percentages, with the condition that incident light interacts with only one surface component. While NLMMs face challenges due to their intrinsic complexity, particularly in modeling and obtaining scene parameters, LMMs remain more widely used in spectral unmixing analysis. Notably, there is a lack of reported studies discussing the validation of spectral unmixing analysis results.

Following shadow detection, efforts to remove or minimize shadow impacts involve algorithms targeting topographic, urban building, cloud, and multi-layered shadows. Recent studies emphasize information recovery from shadows before their elimination, acknowledging the weak information recorded by sensors in shadowed areas [41–43]. Terrain shadows in mountainous regions, caused by low sun elevation angles and steep slopes, reduce reflectance, causing spectral heterogeneity in land cover pixels [44]. Neglecting mountain shadows in forest cover mapping leads to underestimation, emphasizing the need for their removal. Methods like NDVI and band ratios, though simple, are influenced by noise and lose spectral. Utilizing DEM combined with NDVI and topographic correction models are employed to mitigate mountain shadows [45,46]. Urban areas face challenges from building-induced shadows, hindering information extraction from high-resolution images. Recent efforts focus on shadow removal for urban regions, employing algorithms assuming a linear relationship between radiance in shadow and non-shadow areas [47–49]. Shadow detection and de-shadowing are crucial preprocessing steps for satellite images, significantly improving land use and land cover (LULC) classification accuracy and facilitating vegetation carbon density mapping.

Despite challenges in mapping urban vegetation, the integration of methods such as shadow removal and spectral unmixing analysis holds promise for modeling carbon density in urban environments. This research addresses this gap by emphasizing the urgency of developing advanced methods to eliminate the effects of mixed pixels and building and mountain shadows, providing a foundation for precise urban vegetation carbon density estimates. The proposed methodology integrates machine learning algorithms, de-shadowing, and spectral unmixing analysis, offering a comprehensive approach to enhance accuracy in carbon density mapping. The objectives include comparing pixel selection methods, developing shadow removal algorithms, and assessing various spatial modeling techniques. By focusing on Shenzhen, a rapidly growing city in China, this study aims to answer critical questions about model accuracy and shadow removal efficacy. This research contributes to the broader context of combating global warming and underscores the importance of precise information for effective urban vegetation carbon management.

## 2. Study Area and Datasets

### 2.1. Study Area

The study area is situated in Guangdong Province, southern China. Shenzhen city spans from 113°51′ E to 114°21′ E and 22°27′ N to 22°39′ N, covering an area of 1996.85 km$^2$ (Figure 1). Bordered by Mirs Bay to the east, the Pearl River estuary to the west, and adjacent to the New Territory of Hong Kong in the south, Shenzhen features a diverse topography. The southeast exhibits rugged terrain, while the northwest is predominantly flat, with mountainous and hilly areas. Wutong Shan, the highest mountain at 943.7 m, overlooks the coastline extending 230 km with six deep-water ports. The region includes 160 rivulets, major rivers like Shenzhen River, Maozhou River, Longgang River, Guanlan River, and Pingshan River, with catchment areas exceeding 100 km$^2$ but low surface run-off.

Shenzhen experiences a temperate monsoon climate, with a north/northeast monsoon prevailing from September to mid-March, bringing cool, dry air. The summer monsoon dominates from April to September, resulting in hot, humid weather with a heightened risk of typhoons. The annual mean temperature is 22.4 °C, ranging from 12.1 °C in January to 28.1 °C in July, with extremes recorded at 38.7 °C and 0.2 °C. The frost-free period extends over 355 days, with annual mean precipitation of 1933.3 mm mostly occurring between May and September. The city's geology includes granite, naceous shale, tuff, and metamorphic and sandstone rocks. Two main soil types are prevalent: hill soils and alluvial soils, with the latter confined to riverbanks, river plains, and the seashore.

Historically covered by climatic vegetation, Shenzhen's landscape has evolved due to human disturbances. Existing forests, mainly secondary and man-made, encompass lowland evergreen monsoon forests, montane evergreen broad-leaved forests, ravine rainforests, mangroves, and plantations. Since becoming the first special economic zone in 1979, Shenzhen has undergone rapid development, transitioning from a population of 20,000 in 1979 to over 20,709,400 in 2017. This growth has raised environmental concerns, including urban sprawl, diminishing natural resources, reduced forest cover, and pollution. The city's economic progress, while significant, requires a critical examination of its environmental impact and sustainable resource management.

### 2.2. Datasets

In this study, Landsat 8 images from the 8th and 15th of November 2014 were obtained from the United States Geological Survey (USGS) website (http://earthexplorer.usgs.gov/, accessed on 20 February 2018) and served as the primary data source for tasks such as shadow removal, spectral unmixing analysis, model development for estimating vegetation density. Additionally, high-spatial-resolution images from Pleiades 1A and 1B, dated on the 17th, 19th, and 23rd of November 2014, were acquired for land use and land cover (LULC) classification and validation of spectral unmixing analysis. Table 1 provides details on the spectral and spatial resolution characteristics of Landsat 8 datasets and Pleiades-1A and Pleiades-1B datasets.

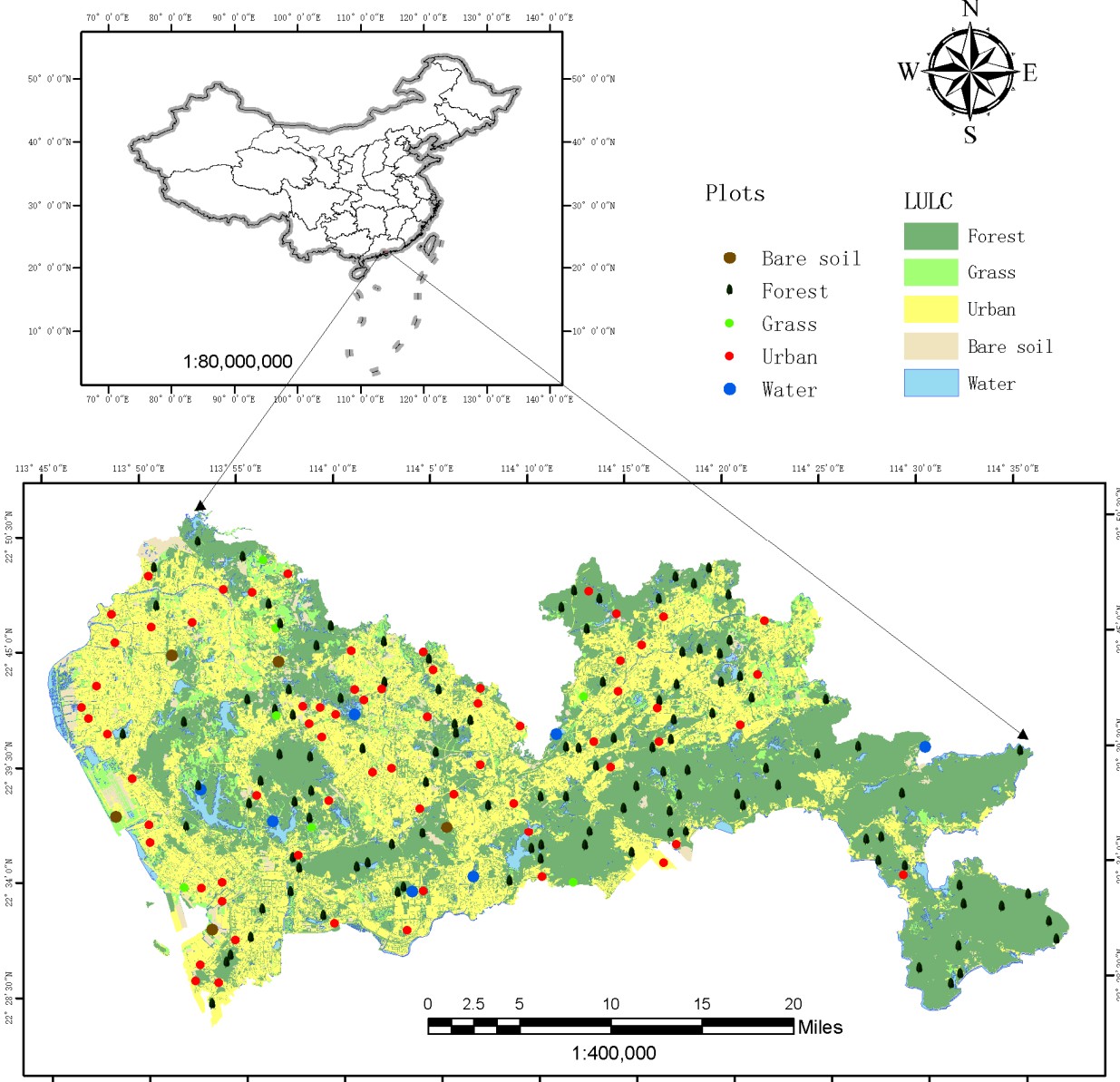

**Figure 1.** The study area, Shenzhen city, and the spatial arrangement of sample plots within land use and land cover (LULC) categories were determined through the implementation of stratified random sampling to establish the plot sets.

The field survey data utilized in this study were gathered between August 2014 and December 2015, aligning with the period of Landsat 8 image acquisition. A total of 188 plots were measured during this timeframe. The sample plot locations were determined using a global positioning system (GPS) device with a positional error of ±5 m. Following the sample plot design specified for the National Forest Inventory in China (Chinese Ministry of Forestry, 1996), variables related to grass, shrub, tree, and stand characteristics were measured. Within each plot (Figure 2), tree attributes, including diameter at breast height (DBH), height, and species, were recorded. For subplots with dimensions of 2 m × 2 m, which contained shrubs and grass, measurements included shrub coverage percentage, ground diameter, height, stock number, and species, as well as grass coverage percentage, species, and height.

**Table 1.** The band information for Landsat 8, Pleiades-1A, and Pleiades-1B.

| Sensor | Band | Range (μm) | Region | Resolution |
|---|---|---|---|---|
| Landsat 8 | Band1 | 0.433–0.453 | Coastal/Aerosol | 30 m |
| | Band2 | 0.450–0.515 | Blue | 30 m |
| | Band3 | 0.525–0.600 | Green | 30 m |
| | Band4 | 0.630–0.680 | Red | 30 m |
| | Band5 | 0.845–0.885 | Near Infrared | 30 m |
| | Band6 | 1.560–1.660 | Short Wavelength Infrared | 30 m |
| | Band7 | 2.100–2.300 | Short Wavelength Infrared | 30 m |
| | Band8 | 0.500–0.680 | Panchromatic | 15 m |
| | Band9 | 1.360–1.390 | Cirrus | 30 m |
| | Band10 | 10.30–11.30 | Long Wavelength Infrared | 100 m |
| | Band11 | 11.50–12.50 | Long Wavelength Infrared | 100 m |
| Pleiades-1A & 1B | Band0 | 0.430–0.550 | Blue | 2 m |
| | Band1 | 0.490–0.610 | Green | 2 m |
| | Band2 | 0.600–0.720 | Red | 2 m |
| | Band3 | 0.750–0.950 | Near Infrared | 2 m |
| | Band4 | 0.480–0.830 | Panchromatic | 0.5 m |

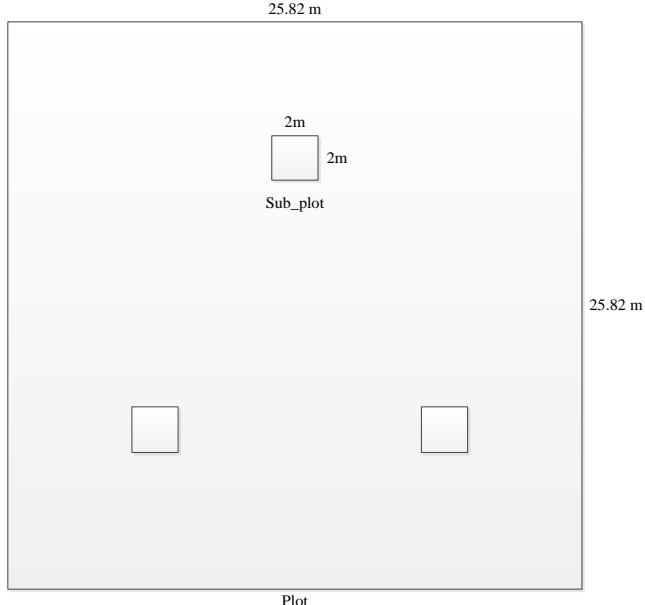

**Figure 2.** Schematic representation of the field plot data collection process for trees (represented by the large square) and shrubs and grass (depicted by three smaller squares).

## 3. Methods

### 3.1. Above-Ground Vegetation Carbon Density Calculation Based on Survey Data

This study employed Pleiades 1A and 1B images for a visual interpretation-based classification of the study area into five LULC types: forests, grasslands, built-up areas, bare lands, and water bodies. The resulting classification guided a stratified random sampling design across the entire study area, ensuring sample sizes proportional to each class's area and random plot locations within each type. For each plot, the biomass values of

trees, shrubs, and grass were individually calculated and subsequently converted into carbon. The total carbon for each plot was determined by summing the biomass values of trees, shrubs, and grass. Subsequently, the plot carbon density was derived based on the respective plot area. The estimation of tree biomass utilized the Tree Volume Calculation Equation, considering both DBH and height measurements for various tree species in Guangdong province (see Appendix A). The obtained tree volume was then converted into biomass and carbon stock pools using conversion coefficients and empirical equations specified for each tree species (NCFCSEM, 2010) as per Equation (1) and Equation (2). Finally, the resulting forest carbon stocks were divided by the plot area to obtain forest carbon density (Mg/ha).

$$BEF \ = \ a + b/M \tag{1}$$

$$B \ = \ BEF * M \tag{2}$$

where BEF represents the biomass expansion factor, which varies across tree species, growing locations, and tree ages. Coefficients 'a' and 'b' are derived from the Biomass and Volume Relationship Parameter Values Table (refer to Appendix B). 'M' represents the volume stock per hectare, and 'B' denotes biomass. The process of converting biomass to carbon is grounded in the Carbon ratio table for different tree species in China (refer to Appendix C).

Regarding shrub and grass carbon estimation, there is a limited number of models or equations available. Fan (2011) proposed two equations, employing a remotely sensed estimation model for estimating shrub and grass biomass. These include the shrub equation Equation (3) and the grass equation Equation (4):

$$Shrub \ bimomass \ = \ 0.0398 \times h_1 - 0.3326 \tag{3}$$

$$Grass \ biomass \ = \ 0.0175 \times h_2 - 0.2888 \tag{4}$$

*3.2. Image Pre-Processing and De-Shadow*

Image pre-processing, involving radiometric and geometric correction, is essential for preparing satellite images for tasks such as LULC classification, spectral unmixing analysis, and model development. Radiometric correction involves standardizing the pixel values of remotely sensed imagery, while geometric correction aligns the imagery to a precise spatial location. For radiometric correction, the Landsat 8 data, initially obtained, underwent calibration to convert digital numbers to at-sensor radiance, ensuring consistency and accuracy in the radiometric values. Additionally, atmospheric correction procedures were applied to mitigate atmospheric interference, enhancing the reliability of spectral information. Geometric correction was conducted using ground control points and Digital Elevation Model (DEM) data to rectify spatial distortions caused by terrain variations. This process ensured accurate alignment of the imagery with the Earth's surface, minimizing geometric errors. These preprocessing steps were vital in mitigating distortions, standardizing radiometric values, and enhancing the overall reliability of the dataset for subsequent analysis. For Landsat 8, Level 2 data was acquired from the USGS service. Regarding Pleiades 1A and 1B images in this study, the process involved initial radiometric calibration using ATCOR 9.5, and the conversion of pixel digital number (DN) values to spectral reflectance was carried out using the Model Maker of ERDAS IMAGINE 2023. Following this, geometric calibration was performed using topographic maps, ensuring a root mean square error (RMSE) less than one pixel, to minimize location errors for Pleiades 1A and 1B images.

The accurate mapping of urban vegetation carbon density is hindered by shadows cast by mountains and tall buildings in urban areas. Therefore, prior to utilizing Landsat 8 images for model development, a shadow removal process was undertaken. Most existing shadow removal approaches operate on the assumption that variations in illumination from different materials can be discerned based on their spectra. In these methods, the

spectral value of a pixel within a shadow, denoted as "y", is treated as a linear combination of endmember spectra from fully illuminated surfaces, $e_i$:

$$y = \sum_i w_{iy} e_i \tag{5}$$

where $w_{iy}$ is the ith endmember weight for the pixel, then assumed that a pure shadow pixel has the reflectance value of 0, the shadow fraction of a mixed pixel can be calculated using equation Equation (6):

$$f_y = \sum_i w_{ia} - \sum_i w_{iy} \tag{6}$$

where $w_{iy}$ is the ith endmember weight for pixel y, the weight can be extracted from each spectrum by taking its dot product with a filter vector that is orthogonal to all endmembers within the pixel, a is a fully illuminated pixel with weights $w_{ia}$ summing to 1, $f_y$ is the shadow fraction of pixel y. This developed a filter used as a mask which could be applied to an image to estimate its shadow fraction $(f_y)$ for each pixel.

In constrained linear spectral unmixing analysis (LSUA), the ith endmember weight could be extracted from each band with a filter vector $v_i$ that is orthogonal to all endmembers; based on this, Equation (7) could be rewritten as:

$$f_y = -\sum_i v_i^T (y - a) = g^T (y - a) \tag{7}$$

where g is a vector used as shadow filter. If the shadow on a scene is rare, the mean of scene spectrum can be used for a fully illuminated pixel (a). And the matched filter could be defined as Equation (8):

$$q = C^{-1}(t - a) / \left[ (t - a)^T C^{-1}(t - a) \right] \tag{8}$$

where q is matched filter, C is covariance matrix, t is the target spectrum. When $t = 0$, the shadow matched filter could be expressed as Equation (9):

$$q_{shadow} = -C^{-1}a / \left( a^T C^{-1}a \right) \tag{9}$$

Application of the matched filter to the Landsat 8 image yields an estimate of pixel-level shadow fraction image (f). Then, the result was rebalanced to simulate illumination by a spectrally uniform source using Equation (10):

$$F(\lambda) = \frac{f(d(\lambda) + s(\lambda))}{fd(\lambda) + s(\lambda)} = f(1 + \frac{s(\lambda)}{d(\lambda)}) / (f + \frac{s(\lambda)}{d(\lambda)}) \tag{10}$$

where $F(\lambda)$ is rebalancing result, $d(\lambda)$ is spectrum of direct sun illumination, $s(\lambda)$ is spectral of sky illumination. After rebalancing, the de-shadow spectrum could be calculated by Equation (11):

$$I = F((\lambda) / (1 - f) \tag{11}$$

In this study, the pure shadow pixels were identified on the Landsat 8 image when using the mosaicked Pleiades 1A and 1B image as a reference.

### 3.3. Spectral Unmixing Analysis

Before linear spectral unmixing analysis (LSUA) was conducted, the Landsat 8 image underwent Minimum Noise Fraction (MNF) transformation to extract noise. Subsequently, a Pixel Purity Index (PPI) was computed using MNF, projecting each pixel onto random vectors in the reflectance space. Pixels scored based on their positions in the projection plot, with the highest scores representing pure pixels. These PPIs were associated with the original image to identify LULC types. N-Dimensional visualization validated endmember purity, eliminating non-corner endmembers for a refined selection.

For spectral unmixing analysis, there is a linear tool in ENVI 4.6, but it is half constrained with the results of unreasonable negative pixel values. In this study, a fully constrained model which solved the negative pixels was developed in Equations (12) and (13) and it guaranteed that the fraction coefficients were positive and their summation was equal to one.

$$\forall i: \ a_i \geq 0 \tag{12}$$

$$\sum_{i=1}^{m} a_i = 1 \tag{13}$$

where $a_i$ represents the fraction of each endmember in pixel $x$.

### 3.4. Modeling

To model vegetation carbon density, 648 spectral variables were derived from the Landsat 8 image (Table 2), encompassing original bands, diverse transformations, band ratios, PCA-generated bands, vegetation indices, difference vegetation indices, and texture measures. Pearson product–moment correlation coefficients were calculated between these variables and vegetation carbon density to assess their significance, utilizing a significance level of 0.05 based on the student's distribution.

**Table 2.** Spectral variables (SVs) obtained from the ten bands of Landsat 8.

| Spectral Variables | Definitions of Spectral Variables | # of SV |
|---|---|---|
| Original band | Band 1 (Coastal Aerosol), Band 2 (Blue), Band 3 (Green—GRN), Band 4 (Red), Band 5 (Near Infrared—NIR), Band 6 (Shortwave Infrared 1—SWIR1), Band 7 (Shortwave Infrared 2—SWIR2), Band 8 (Cirrus), Band 9 (Long Wavelength), and Band 10 (Long Wavelength) | 10 |
| Inversions of bands | $IB_i = \frac{1}{band_i}$ , $i = 1, \ldots 10$ | 10 |
| Simple two-band ratios | $SR_{i,j} = \frac{Band_i}{Band_j}$, $i, j = 1, \ldots 10; i \neq j$ | 90 |
| Three-band ratios | $TR_{i,j,k} = \frac{Band_i}{Band_j + Band_k}$ , $i, j, k = 1, \ldots 8; j \neq j \neq k$ | 359 |
| Difference vegetation indices | $DVI_{i,j} = Band_i - Band_j \ i, j = 1, \ldots 10; i \neq j$ | 45 |
| Shortwave infrared-visible band ratio | $SVR = SWIR1 / \left[ \frac{RED + GRN}{2} \right]$ | 1 |
| Normalized difference vegetation index | $NDVI = \frac{NIR - RED}{NIR + RED}$ | 1 |
| Modified normalized difference vegetation index | $MNDVI = \frac{NIR - RED}{NIR + RED} \left( 1 - \frac{SWIR1 - SWIR1_{min}}{SWIR1_{max} - SWIR1_{min}} \right)$ | 1 |
| Red–green vegetation index | $GRVI = (RED - GRN) / (RED + GRN)$ | 1 |
| Reduced simple ratio | $RSR = \frac{NIR}{RED} \left( 1 - \left( \frac{SWIR1 - SWIR1_{min}}{SWIR1_{max} - SWIR1_{min}} \right) \right)$ | 1 |
| Soil adjusted vegetation index | $SAVI_l = \frac{(NIR - RED)(1+l)}{NIR + RED + l}$, $l = 0.1, 0.25, 0.3, 0.5$ | 4 |
| Atmospherically resistant vegetation index | $ARVI = [NIR - (2 \times RED - BLUE)] / [NIR + (2 \times RED - BLUE)]$ | 1 |
| Enhanced vegetation index | $EVI = 2.5(NIR - RED) / (NIR + 6RED - 7BLUE + 1)$ | 1 |
| Principal component analysis | The first 3 PCs from Principal component analysis (PCA) | 3 |
| Texture measures | Texture measures derived from the Grey-Level Co-occurrence Matrix, encompassing mean, angular second moment, contrast, correlation, dissimilarity, entropy, homogeneity, and variance. | 80 |

Spatial autocorrelation analysis revealed significant clustering of plot carbon density values across the study area, indicating varying relationships between vegetation carbon values and original Landsat 8 spectral bands in different locations. This variability and nonlinearity are evident in scatter plots shown in Figure 3.

Considering the intricate urban landscape and the nonlinear relationship between vegetation carbon density and Landsat 8 images' original bands (Figure 3), this study employed two global parametric spatial interpolation models—Linear Stepwise Regression (LSR) and Logistical Model based Stepwise Regression (LMSR) [50–54]—along with three local non-parametric models—k-Nearest Neighbors (kNN) [55], Decision Trees (DT) [56], and Random Forests (RF) [57]. These models were utilized to map vegetation carbon density, incorporating spectral variables and vegetation fraction derived from LSUA, in comparison to the complex urban context.

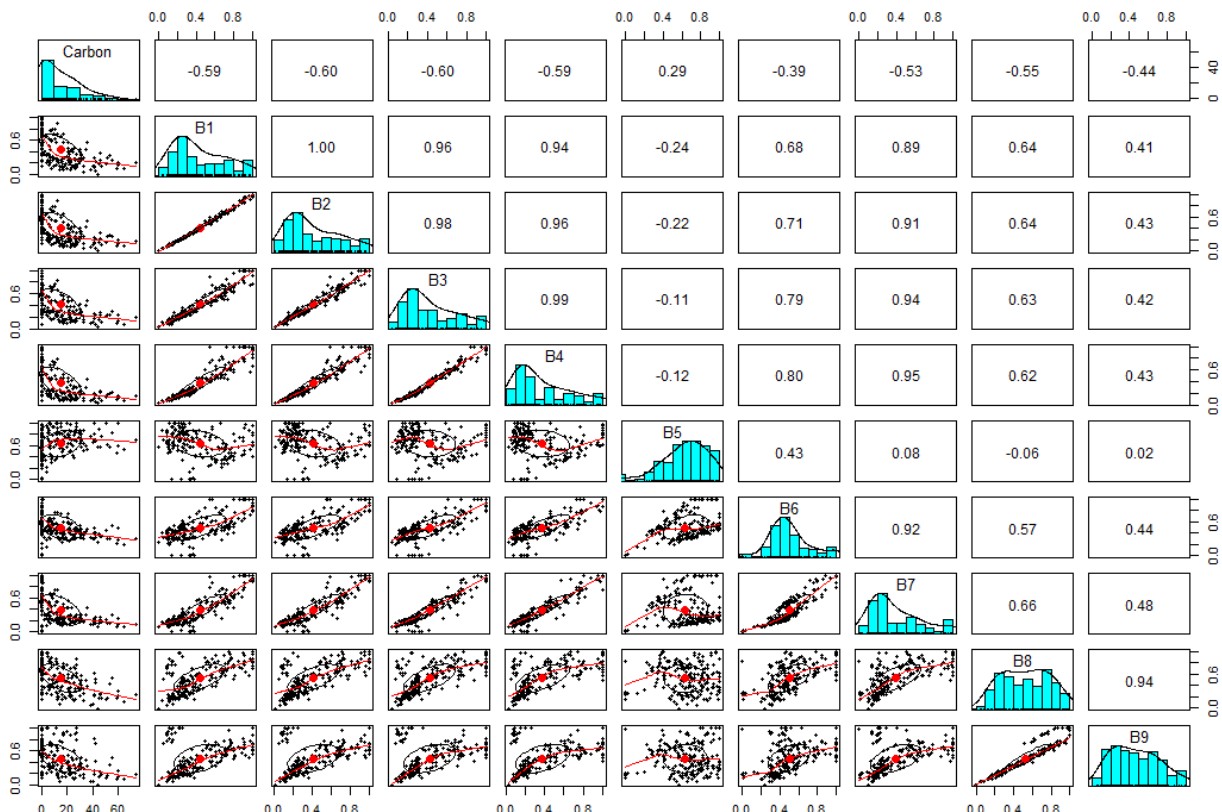

**Figure 3.** Correlation coefficient analysis examining the relationship between vegetation carbon density and the Landsat 8 images' original bands. The symbols in the figure denoted as Carbon, B1, B2, . . ., B9 correspond to vegetation carbon density, Landsat Band 1, Band 2, . . ., Band 9, respectively.

### 3.4.1. Linear Stepwise Regression Model

The LSR model, commonly applied in mapping forest biomass and carbon density, was employed in this study to identify statistically significant spectral variables that enhance model fit and reduce sum of squared errors. Additionally, it assessed collinearity among spectral variables from Landsat 8 imagery, and the chosen significant independent variables were subsequently utilized in the backward-based linear stepwise regression, Equation (14)

$$y = \beta_0 + \beta_1 X_1 + \cdots + \beta_i X_i + \cdots + \beta_n X_n \tag{14}$$

where y is the vegetation carbon density, $\beta_i$ is the *i*th coefficient and $X_i$ is the ith spectral variable derived from the Landsat 8 image.

### 3.4.2. Logistical Model Based Stepwise Regression Model

The LMSR, a probabilistic statistical prediction model handling binary dependent variables, was employed in this study with a dependent variable ranging from 0 to 1. Standardizing the plot vegetation carbon density values to this range, LMSR, coupled with stepwise regression, identified significant spectral variables. Multicollinearity analysis, utilizing the variance inflation factor (VIF), flagged highly correlated variables (coefficients ranging from 0.03 to 0.99). Variables with VIF exceeding 10 were considered indicative of severe multicollinearity. The stepwise logistic regression, conducted in R statistical software, version 3.5.3, employed the LMSR model, denoted by Equation (15).

$$P = \frac{e^{b_0 + b_1 x_1 + b_2 x_2 + \cdots + b_n x_n}}{1 + e^{b_0 + b_1 x_1 + b_2 x_2 + \cdots + b_n x_n}} \tag{15}$$

where $P$ represents standardized plot vegetation carbon density values, $e$ is the natural logarithm base, $b_0$ is the interception at $y$-axis and $b_i$ is the coefficient of the $i$th independent variable $x_i$, $x_i$ is the $i$th significant spectral variable derived from the Landsat 8 image.

### 3.4.3. k Nearest Neightbors

The kNN is a simple, intuitive, and nonparametric method in statistical discrimination, used for classification or regression. It predicts unknown attributes based on the observed learning set and relies on distance measures. Formally, the model can be described as follows:

Let $L = \{(y_i, x_i), i = 1, 2, 3, \ldots, n_L)\}$ be observed data and used as training dataset, $y_i$ denotes class membership, and $x_i$ represents the predictor variables. For a new observation $(y, x)$, the nearest neighbor $(y_{(1)}, x_1)$ is determined by an arbitrary distance function as Equation (16):

$$d\left(x, x_{(1)}\right) = min_i(d(x, x_i)) \tag{16}$$

And $\hat{y} = y_{(1)}$, the nearest neighbor is selected as prediction for $y$. Classically, distance functions are Euclidean distance or absolute distance.

The idea of using multiple closest observations within the learning set was extended, leading to the k-nearest neighbors method (kNN). Users can set the number of nearest neighbors, k, and consider distance weight in the model. The kNN model assumes that closer neighbors have higher influence on the decision. However, in this study, all k nearest neighbors were assumed to have equal influence. Before searching for the nearest neighbors, similarity measures needed to be standardized for use as weights.

The kNN model is widely used for predicting forest attributes, biomass, and carbon density. It estimates the values of an interest variable based on the similarity of predictor variables with k nearest neighbors or selected plots. In this research, the urban vegetation carbon density at each location was estimated by weighting the carbon density values of the nearest plots using the inverses of Euclidean distances. The variable distance was weighted by triangular, rectangular, Epanechnikov, Gaussian, rank, and optimal kernel functions.

Rectangular k

ernel : $\frac{1}{2} * I(|d| \leq 1)$

Triangular kernel : $|(1 - (d)) * I(|d| \leq 1)$

Epanechnikov kernel : $\frac{3}{4}\left(1 - d^2\right) * I(|d| \leq 1)$

Gaussian kernel : $\frac{1}{\sqrt{2\pi}}\exp\left(-\frac{d^2}{2}\right)$

Rank kernel : $1/d$

Optimal kernel : $\frac{35}{32}\left(1 - d^2\right)^3 * I(|d| \leq 1)$

where $d$ is distance, $I(.)$ is indicator function: if defined condition in brackets is true, $I(.) = 1$ and otherwise, $I(.) = 0$. The window width of kernel function was determined by a certain distance from maximum value. The Euclidean distances were standardized by dividing itself using the closest neighbor (k + 1) that was not used for predication.

The used significant spectral variables were derived from both LSR and LMSR. As a first step, the spectral variables were standardized by dividing themselves using their standard deviation.

### 3.4.4. Decision Trees

The DT, introduced by Breiman (1984), is employed for classification or regression predictive analysis. Also known as "decision trees" or CART in some contexts, this algorithm forms the basis for important algorithms like Random Forests [58]. Constructing a DT involves selecting input variables and split points using a greedy algorithm to minimize a cost function. A predefined stopping criterion is essential to avoid an infinite model run.

In regression predictive modeling, the cost function minimization determines split points using the sum squared error calculated from training samples, Equation (17)

$$SQE \ = \ sum\left(y - \hat{y}\right)^2 \tag{17}$$

where *SQE* is sum squared error, *y* is output of the training sample, and $\hat{y}$ is predicted output.

The stopping criterion is crucial in DT, commonly set as a minimum count based on the number of training instances for each leaf node. When the count falls below this minimum, the split stops, and the node becomes a final leaf node. This criterion significantly impacts DT performance. The complexity of DT is linked to the number of splits; simpler trees are preferred to prevent overfitting. Pruning the tree to minimize cross-validation error is necessary for avoiding overfitting issues.

### 3.4.5. Random Forests

RF was developed by Breiman (2001), enhances categorical variable-based classification and continuous variable-based regression trees (CART) by amalgamating multiple sets of decision trees [59]. As an effective ensemble machine learning model, RF is adept at classification or regression predictive analytics, employing an additive model to combine decisions from a sequence of base models, Equation (18)

$$g(x) \ = \ f_0(x) + f_1(x) + f_2(x) + f_3(x) + \cdots \tag{18}$$

where $g(x)$ is the sum of simple based models $f_i$, in this study, each base classifier is a simple decision tree used for regression prediction. For Random Forest regression, all the base models are trained independently using different subsamples of observations. Each tree node splitting is based on a deterministic algorithm by randomly selecting a sub-set of variables and a sample from the training data [59]. In this study, RF begins with randomly drawing many bootstrap sub-samples with replacement form the field plot observations. A regression tree was constructed for each sub-sample. For nodes of each tree, a small part of input variables was selected from the total inputs used for binary partition. For the tree splitting criterion, it was based on the lowest Gini Index (Equation (19)) of the chosen input variable.

$$I_g \ = \ 1 - \sum_{j=1}^{m} f\left(t_{X(x_i)}, j\right)^2 \tag{19}$$

where, $f\left(t_{X(x_i)}, j\right)$ is the section of samples which the value $x_i$ belongs to the leave *j* as node *t*. The predicted carbon density at a location without observance was calculated by averaging the bootstrap selected sub-samples constructing trees.

In the construction of the RF, two parameters must be optimized: the number of trees (ntree) and the optimal minimal size of the terminal nodes of the trees. The optimal numbers of trees and nodes for predicting vegetation carbon density were determined based on the root mean square error (RMSE) of calibration.

### 3.5. Accuracy Assessment

In comparison, all models employed were combined with LSUA to validate our hypothesis: whether the addition of the vegetation fraction image from LSUA improves the models' performance in predicting carbon density. The accuracy assessment of vegetation carbon density estimates utilized a cross-validation method. In this procedure, a random sample of plot vegetation carbon density was first removed from the field plots, with the remaining plots utilized to train the models. The estimate of vegetation carbon density for the removed sample was calculated, and the deviation between the estimated and observed values was obtained. Another sample was then randomly removed, and the previously removed one was reintroduced into the dataset. The corresponding modeling and estimation were conducted for this sample. This process was repeated until all samples were estimated. Based on the predictions and their comparison with field measurements,

the coefficient of determination $R^2$ and root mean square error (RMSE) were derived to assess the goodness-of-fit and prediction performance of the models.

## 4. Results

### 4.1. Statistics of Field Data

Table 3 presents a statistical summary of the observed plot data employed in mapping urban vegetation carbon density in Shenzhen. The sample mean and standard deviation at the plot level were estimated using simple random sampling estimators. The plot vegetation carbon density values exhibited a substantial coefficient of variation, with a confidence interval ranging from 12.66 Mg/ha to 17.32 Mg/ha at a significance level of 0.05.

**Table 3.** Statistical summary of sample plot data utilized for urban vegetation carbon density mapping.

| Number of Plots | Minimum (Mg/ha) | Maximum (Mg/ha) | Sample Mean (Mg/ha) | Standard Deviation (Mg/ha) | Coefficient of Variation (%) |
|---|---|---|---|---|---|
| 188 | 0 | 73.550 | 14.99 | 16.3 | 108.87 |

### 4.2. Correlation of Vegetation Carbon Density with Spectral Variables

The Pearson product–moment correlation coefficients between observed plot vegetation carbon density and 648 spectral variables were calculated. Prior to shadow removal, 523 spectral variables exhibited coefficients ranging from 0.142 to 0.667, significantly different from zero at a 0.05 significance level. After shadow removal, the number of significant variables increased to 534, with coefficients ranging from 0.146 to 0.688. Shadow removal positively impacted the correlation coefficient between field observations and spectral variables. The band-ratio TR536 showed the highest correlation with vegetation carbon density both before and after shadow removal of Landsat 8 imagery. Following shadow removal, Table 4 lists the original Landsat 8 image bands and 45 other spectral variables with the highest correlations to plot carbon density observations. In comparison to vegetation indices and band ratios, PCAs and matrix-based texture variables exhibited smaller correlation coefficients.

**Table 4.** Pearson correlation coefficients (r) were calculated between field-measured carbon density and spectral variables (top 45 highest correlated variables and 9 original Landsat bands) after shadow removal (n = 188; veg-fraction: vegetation fraction obtained using LSUA).

| Spectral Variables | Correlation | | Spectral Variables | Correlation | |
|---|---|---|---|---|---|
| | r | P | | r | P |
| B1 | −0.593 | 0 | TR415 | −0.661 | 0 |
| B2 | −0.596 | 0 | TR416 | −0.608 | 0 |
| B3 | −0.597 | 0 | TR425 | −0.660 | 0 |
| B4 | −0.586 | 0 | TR426 | −0.596 | 0 |
| B5 | 0.293 | $4.44 \times 10^{-5}$ | TR435 | −0.650 | 0 |
| B6 | −0.394 | $2.23 \times 10^{-8}$ | TR436 | −0.574 | 0 |
| B7 | −0.529 | $5.77 \times 10^{-15}$ | TR458 | −0.580 | 0 |
| B9 | −0.554 | 0 | TR459 | −0.570 | 0 |
| B10 | −0.435 | $4.42 \times 10^{-10}$ | TR516 | 0.658 | 0 |
| DVI56 | 0.642 | 0 | TR517 | 0.612 | 0 |
| DVI57 | 0.617 | 0 | TR526 | 0.669 | 0 |
| ARVI | 0.626 | 0 | TR527 | 0.631 | 0 |
| MNDVI | 0.630 | 0 | TR534 | 0.581 | 0 |
| SAVI0.1 | 0.631 | 0 | TR536 | 0.688 | 0 |
| SAVI0.25 | 0.629 | 0 | TR537 | 0.660 | 0 |
| SAVI0.5 | 0.627 | 0 | TR546 | 0.685 | 0 |

**Table 4.** *Cont.*

| Spectral Variables | Correlation | | Spectral Variables | Correlation | |
|---|---|---|---|---|---|
| | r | P | | r | P |
| SR57 | 0.686 | 0 | TR547 | 0.654 | 0 |
| SR67 | 0.639 | 0 | TR567 | 0.684 | 0 |
| TR125 | −0.584 | 0 | TR637 | 0.583 | 0 |
| TR135 | −0.543 | $8.88 \times 10^{-16}$ | TR647 | 0.592 | 0 |
| TR215 | −0.621 | 0 | TR715 | −0.531 | $4.88 \times 10^{-15}$ |
| TR235 | −0.569 | 0 | TR725 | −0.531 | $4.44 \times 10^{-15}$ |
| TR258 | −0.511 | $6.33 \times 10^{-14}$ | TR735 | −0.527 | $8.44 \times 10^{-15}$ |
| TR315 | −0.650 | 0 | TR745 | −0.526 | $8.88 \times 10^{-15}$ |
| TR325 | −0.641 | 0 | TR758 | −0.510 | $7.82 \times 10^{-14}$ |
| TR345 | −0.561 | 0 | TR759 | −0.524 | $1.24 \times 10^{-14}$ |
| TR358 | −0.516 | $3.46 \times 10^{-14}$ | Veg_fraction | 0.595 | 0 |

*4.3. Spectral Unmixng Analysis*

For spectral unmixing analysis, two endmember selection methods were compared: spectral characteristics-based automatic selection and operator's knowledge-based manual selection. Additionally, three sets of endmembers (2, 3, and 4) were evaluated: vegetation, urban; vegetation, urban, water; and vegetation, urban, water, bare soil. Results demonstrated that, after shadow removal, the 4-endmember configuration, whether automatically or manually selected, produced the highest correlation coefficient of 0.595 between the vegetation fractional images and field plot carbon density (Table 5). The correlation coefficient increased with the number of endmembers, reaching a peak at 4 endmembers, after which it began to decrease.

**Table 5.** Summary statistics of correlation coefficients between vegetation fractional images and field-measured carbon density for different endmember configurations (2 endmembers: vegetation, urban; 3 endmembers: vegetation, urban, water; and 4 endmembers: vegetation, urban, water, bare soil) both before and after shadow removal.

| Method | 2-Endmember | 3-Endmember | 4-Endmember |
|---|---|---|---|
| Automatical selection (Before) | 0.491 | 0.554 | 0.589 |
| Manual selection (Before) | 0.492 | 0.555 | 0.59 |
| Automatical selection (After) | 0.495 | 0.563 | 0.595 |
| Manual selection (After) | 0.498 | 0.564 | 0.595 |

Given the superior performance of the automatic endmember selection method over manual selection, the study employed the results from decomposing mixed pixels using 4 endmembers with automatic pure pixel selection for model development (Figure 4).

Figure 4 illustrates the predominant distribution of vegetation in Shenzhen city's southeast, southwest central, northeast, and northwest regions, where urban fraction estimates were relatively smaller. The linear model effectively identified water bodies, discerning their vegetation and urban fractions approaching zero. This model exhibited a notable correlation coefficient of 0.595 with plot vegetation carbon density.

Validation of the vegetation fraction image was conducted using a visual interpretation map of Land Use and Land Cover (LULC) derived from Pleiades 1A and 1B images with a spatial resolution of 0.5 m. When compared with the vegetation fraction image (Figure 4a), spectral unmixing analysis successfully extracted detailed vegetation cover information. The fully constrained linear spectral unmixing analysis accurately estimated both the spatial pattern and specific coverage rate of urban vegetation. The resulting vegetation cover percentage for the entire study area was 44.2% based on Pleiades image visual interpretation and 41.7% for the fully constrained linear spectral unmixing analysis using Landsat 8 imagery. The pixel-based Root Mean Square Error (RMSE) was 0.16.

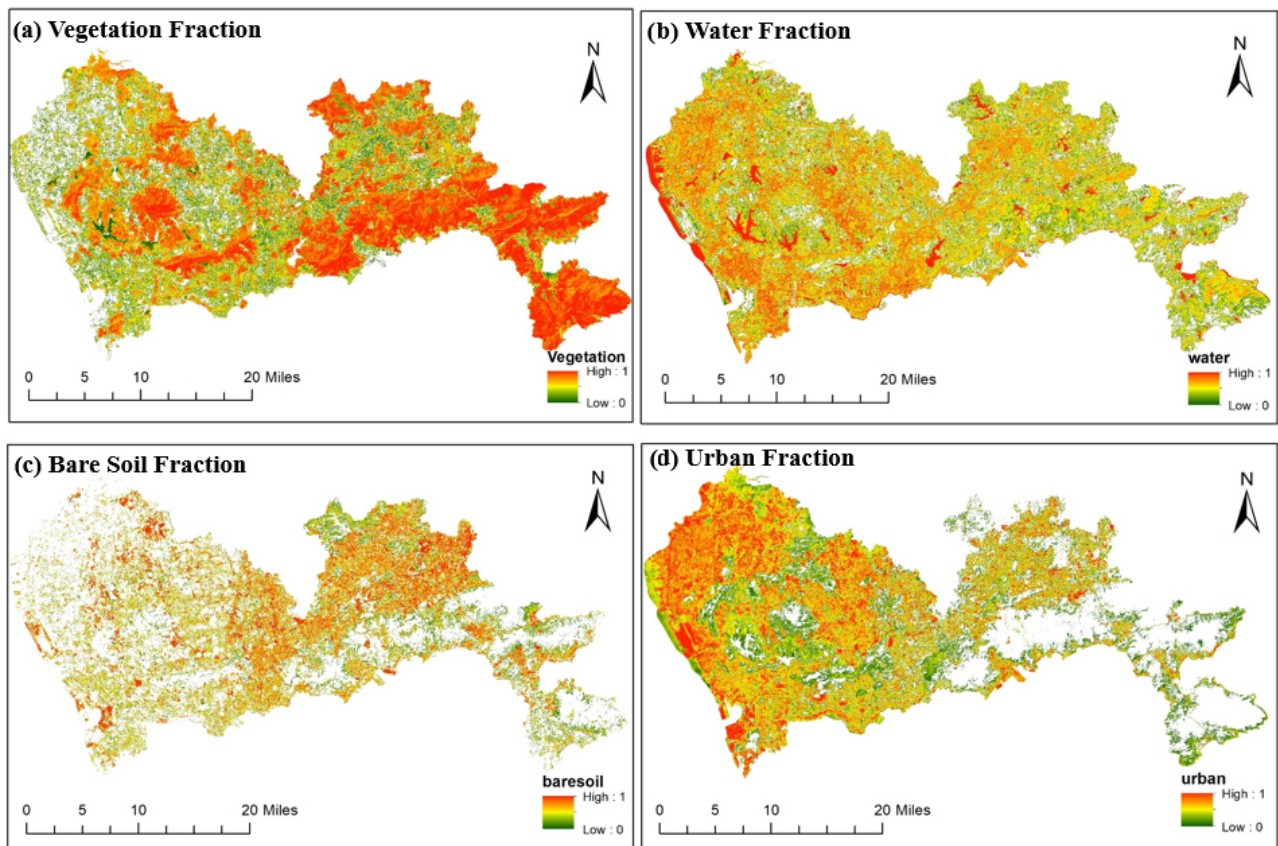

**Figure 4.** The fractional images were acquired through a mathematical selection method for decomposing mixed pixels utilizing 4 endmembers.

*4.4. De-Shadow Results of Landsat 8 Image*

Shadow removal, facilitated by LSUA, involved selecting pure endmembers for shadow, vegetation, urban, water, and bare soil in the spectral unmixing analysis. A shadow fraction image served as a mask layer for shadow removal based on each pixel's shadow contribution. Figure 5 displays the image before shadow removal, revealing identifiable mountain shadows. Given the Landsat 8 image's 30 m × 30 m spatial resolution, building shadows were challenging to discern, making it difficult to assess the shadow removal effect. The accuracy assessment utilized 300 random sample points, categorizing them as poor, average, or good based on their visual representation of original land cover types compared with the pre-shadow removal Landsat 8 image, with assistance from Pleiades 1A and 1B images as a reference.

Among the 300 pixels, 29 were located in mountain-shadowed areas, where shadows were successfully removed or alleviated, leading to a significant recovery of LULC information. However, in urban areas, only a few pixels could be clearly identified as shadows with the assistance of Pleiades 1A and 1B. The method performed well in shadow removal in mountainous areas but faced challenges in urban areas due to coarse resolution.

Comparing Figure 5 results with Pleiades 1A and 1B as a reference, mountain shadows were effectively removed, while shadows induced by buildings showed a fair improvement in image quality. Table 6 presents a correlation coefficient comparison between plot vegetation carbon density and spectral variables using pre- and post-shadow-removed Landsat 8 images. After shadow removal, correlation coefficients increased by 1.28% to 2.59% across different bands. Given the positive impact of the shadow removal algorithm on all bands, all models were constructed using the post-shadow-removed Landsat 8 image.

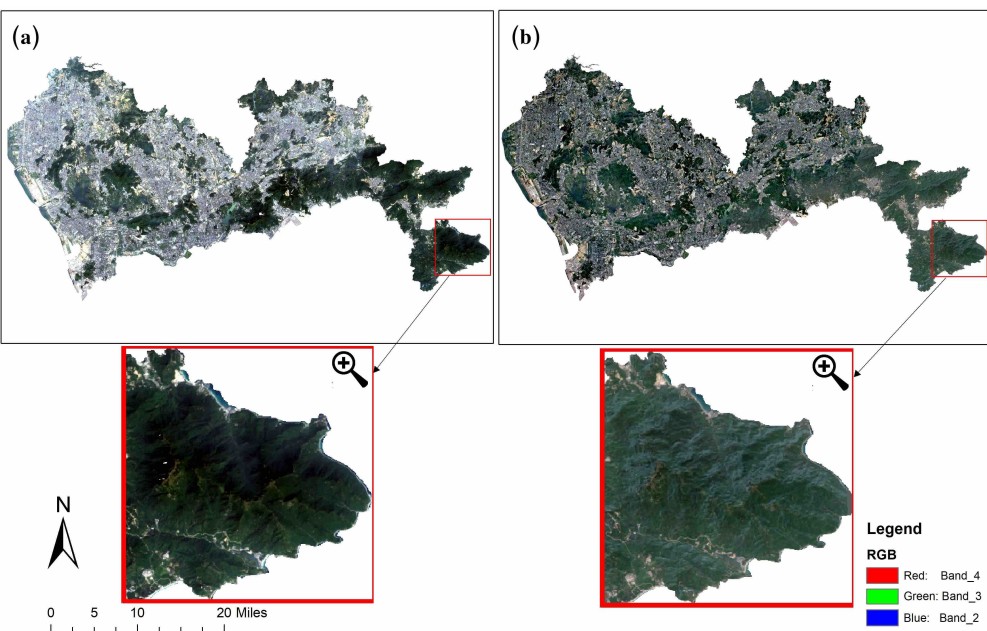

**Figure 5.** Comparison between pre- and post-shadow removal of Landsat 8: (**a**) Landsat 8 image before shadow removal displayed in natural color; (**b**) Landsat 8 image after shadow removal displayed in natural color. The red highlighted area is enlarged to illustrate the differences before and after shadow removal.

**Table 6.** Correlation coefficients between vegetation carbon density and spectral variables using Landsat 8 images before and after shadow removal.

| Landsat 8. | B1 | B2 | B3 | B4 | B5 | B6 | B7 | B9 | B10 |
|---|---|---|---|---|---|---|---|---|---|
| Before | −0.578 | −0.581 | −0.587 | −0.571 | 0.283 | −0.389 | −0.518 | −0.546 | −0.423 |
| After | −0.593 | −0.596 | −0.597 | −0.586 | 0.294 | −0.394 | −0.529 | −0.554 | −0.435 |

*4.5. Vegetation Carbon Density Mapping*

The performance of LSR models with and without the vegetation fraction image was compared (Figure 6a,b). Four stepwise-selected spectral variables (excluding the vegetation fraction variable) were used as independent variables to fit the LSR model for estimating vegetation carbon density (Equation (20)). The variable TR536, which exhibited the highest correlation with plot vegetation carbon density, was excluded by stepwise regression due to its high collinearity with other selected variables.

$$\hat{Y} = 24.976 − 14.968 * B9 + 44.513 * TR567 − 25.550 * SR67 − 3.753 * B2\_mean \quad (20)$$

Vegetation carbon density estimates using Equation (20) ranged from −77.283 Mg/ha to 454.69 Mg/ha, with a mean estimate of 15.332 Mg/ha, slightly overestimating the observed mean of 14.999 Mg/ha. Leave-one-out cross-validation resulted in an $R^2$ of 0.5451 and RMSE of 10.852 Mg/ha (Table 7).

Upon integrating the vegetation fractional image into the LSR model, the selected variables changed, and the model was expressed as Equation (21). This integration led to a slight improvement, with $R^2$ increasing to 0.5453, RMSE decreasing to 10.812, and the mean estimation closer to field measurement compared to the LSR model alone (Table 7). Figure 6 illustrates the enhanced estimation of sub-pixel carbon density in the east urban areas along the roads when combining LSR and LSUA, reducing the absolute values of negative estimates from LSR.

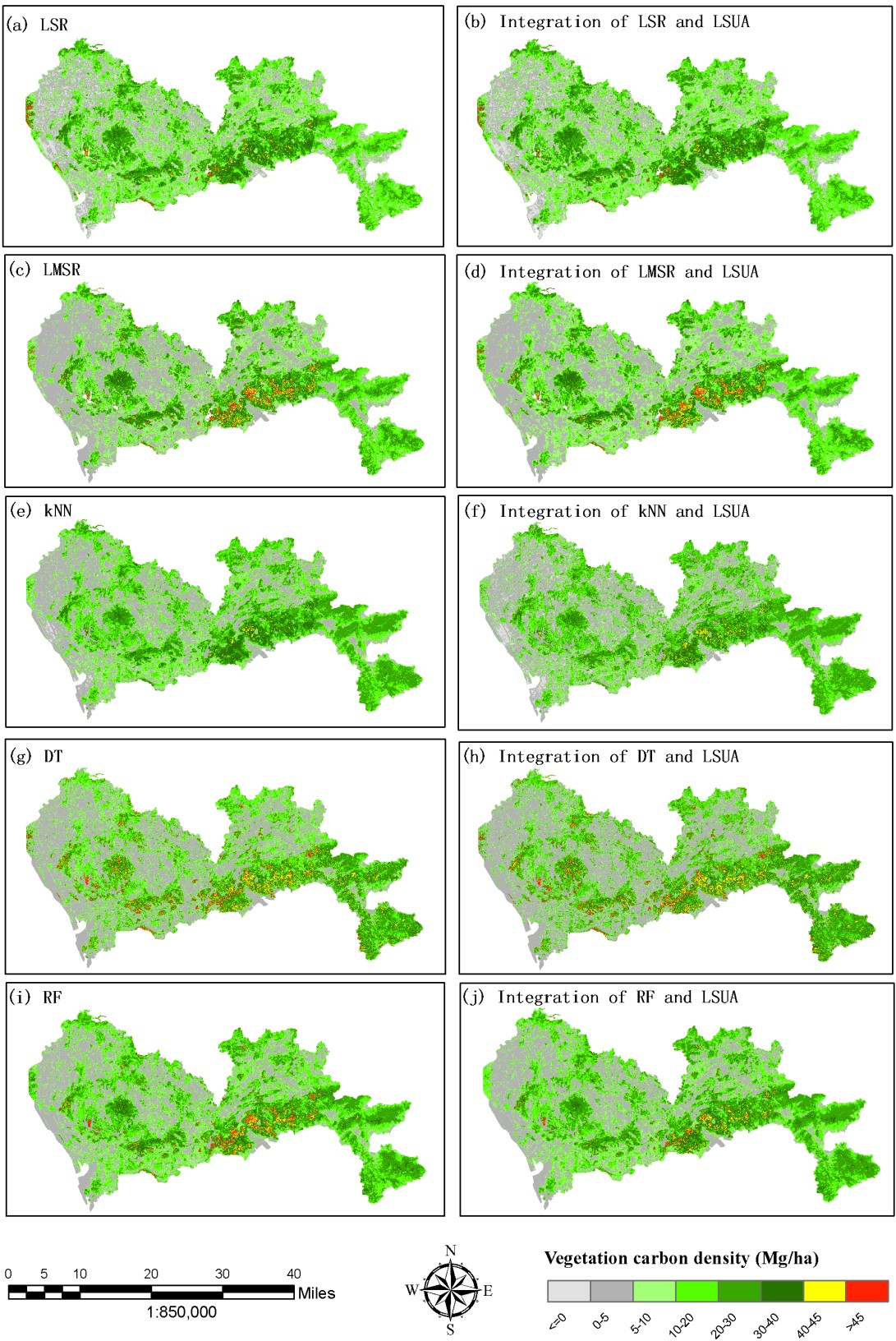

**Figure 6.** Comparison of the mapping outcomes for vegetation carbon density utilizing LSR, LMSR, kNN, DT, and RF models with and without the incorporation of the vegetation fraction variable derived from LSUA in the modeling process.

**Table 7.** The accuracy assessment of vegetation carbon density estimates from LSR, LMSR, kNN, DT, RF models and these models integrated with LSUA. In the table, Mean indicates observed and model predication mean of vegetation carbon density, R$^2$ is the coefficient of determination and RMSE is mean square error. û is map mean estimate and Varmap is the variance of the map based on model-assisted regression estimators.

| Approach | Mean | R$^2$ | RMSE | $\hat{\mu}$ (Mg/ha) | Var$_{map}$ |
|----------|------|-------|------|-------------------|-------------|
| Observed | 14.99 | - | - | - | - |
| LSR | 15.07 | 0.5451 | 10.852 | 15.332 | 1.68 |
| LSR integrated with LSUA | 15.05 | 0.5453 | 10.812 | 15.26 | 1.61 |
| LMSR | 14.91 | 0.5621 | 9.153 | 14.091 | 1.38 |
| LMSR integrated with LSUA | 14.94 | 0.5712 | 9.046 | 14.256 | 1.34 |
| kNN | 14.75 | 0.4620 | 10.561 | 14.483 | 1.89 |
| kNN integrated with LSUA | 14.86 | 0.4641 | 9.682 | 14.518 | 1.77 |
| DT | 15.00 | 0.8171 | 6.952 | 14.501 | 1.26 |
| DT integrated with LSUS | 15.00 | 0.8205 | 6.888 | 14.501 | 1.24 |
| RF | 15.33 | 0.7630 | 8.741 | 15.419 | 1.16 |
| RF integrated with LSUA | 15.20 | 0.7800 | 8.651 | 15.136 | 1.09 |

$$\hat{Y} = 33.274 - 17.406 * B9 - 14.294 * TR426 + 42.448 * TR567 - 27.295 * SR67 + 2.280 * Veg\_fraction \tag{21}$$

The non-linear LMSR model, utilizing two significant spectral variables (B9 and TR567) selected by stepwise regression (Equation (22)), demonstrated a more reasonable estimate range from 0 Mg/ha to 73.555 Mg/ha compared to the LSR model. The LMSR achieved an R$^2$ of 0.5621, surpassing both LSR models with and without LSUA integration (Table 7). Additionally, the RMSE was reduced to 9.153 Mg/ha, smaller than those obtained with LSR alone or with LSUA integration.

$$\hat{Y} = \frac{73.555 * \exp(-2.6678 - 1.3674 * B9 + 1.9713 * TR567)}{1 + \exp(-2.6678 - 1.3674 * B9 + 1.9713 * TR567)} \tag{22}$$

When the LMSR model integrated with LSUA (Equation (22)), the coefficient of determination increased to 0.571, surpassing the LMSR model without the inclusion of the vegetation fraction variable. Simultaneously, the RMSE decreased to 9.046 Mg/ha, representing a 1.2% reduction compared to LMSR without the vegetation fraction variable integration (Table 7).

Figure 6c,d illustrates the ability of the LMSR model, with and without the vegetation fraction variable, to capture the spatial patterns of vegetation carbon density across the study area. Areas marked in grey signify low vegetation carbon density, predominantly in developed urban regions. The LMSR model, combined with LSUA, produces a more reasonable sub-pixel vegetation carbon density map compared to the LSR model. In mountainous and urban park areas, carbon density falls within the 20 Mg/ha to 30 Mg/ha range, while values of 30 Mg/ha to 80 Mg/ha are primarily found in low-elevation mountainous areas with favorable soil conditions, lower slopes, and more suitable temperatures than high-elevation counterparts.

For the kNN model, optimal parameters such as the number of nearest neighbors (k), spectral distance parameter, weighting kernel function, and predictor set were determined through iterative kNN imputation and mean square error analysis. The plot vegetation carbon density served as the dependent variable, while the significant variables selected for LSR and LMSR were used as independent variables. For kNN without LSUA, the optimal k was 12, and the best distance weighting kernel function was rectangular, resulting in the smallest mean squared error of 0.0247. With the integration of LSUA, the optimal k remained 10, and the best distance weighting kernel function was also rectangular, resulting in the smallest mean squared error of 0.0244.

For kNN integrated with LSUA, the coefficient of determination was 0.4641, surpassing that of kNN without the vegetation fraction variable. The RMSE decreased to 9.682 Mg/ha, showing an 8.79% reduction compared to kNN without the vegetation fraction variable (Table 7). The map variance decreased by 6.34%, and the estimated sample mean and map mean increased, aligning more closely with the field measurement mean compared to kNN without the vegetation fraction variable. However, both models exhibited overestimation in areas with low values and underestimation in areas with high vegetation carbon density.

Both kNN and kNN with LSUA tended to underestimate vegetation carbon density in areas with values exceeding 30 Mg/ha and overestimate it in areas with values below 20 Mg/ha, Figure 6e,f. Both models effectively captured the overall patterns of vegetation carbon density in the urban area. The integration of kNN with LSUA performed significantly better in urban areas, particularly in extracting subpixel vegetation carbon information for roadside trees.

The DT is commonly employed for classification and regression tasks. Figure 6g,h depict the DT model's estimation of vegetation carbon density. For both the DT and the DT integrated with LSUA, the flowcharts progress downward from the top, starting with predictor TR567, which exhibits the highest correlation with plot vegetation carbon density among all independent variables in the DT model. Based on the criterion of a TR567 value less than 0.8 or not, the 188 plots are divided into two groups: 90 plots with TR567 less than 0.8 and the remaining plots with TR567 not less than 0.8. This process continues until reaching a final leaf node. The number of splits serves as a crucial indicator for evaluating the effectiveness of the DT model.

With field measurements incorporated into the DT model and a minimum split number of six, both the DT and DT integrated with LSUA achieved the highest coefficient of determination ($R^2$) between observed and predicted vegetation carbon density values. The DT resulted in an $R^2$ of 0.8171, with plot and map mean estimates of 15.00 Mg/ha and 14.501 Mg/ha, respectively, and an RMSE of 6.952 Mg/ha. Comparatively, the integration of DT with LSUA showed slight improvement, with an $R^2$ of 0.8205, plot and map mean of 15.00 Mg/ha and 14.501 Mg/ha, respectively, and an RMSE of 6.888 Mg/ha (Table 7).

Both the DT model and the DT integrated with LSUA demonstrated significant predictive capability for urban vegetation carbon density. Given the potential overfitting concern associated with decision trees, the absence of end nodes with n values of 1 or 2 in this study (Figure 6g,h) suggests reliable model performance.

For the optimal number of trees in RF, RMSE showed a decreasing trend with an increasing number of trees (Figure 7, left). Before integrating RF with LSUA, the error stabilized after the number of trees exceeded 500. Upon integration with LSUA, the error stabilized when the number of trees exceeded 800 (Figure 7, right).

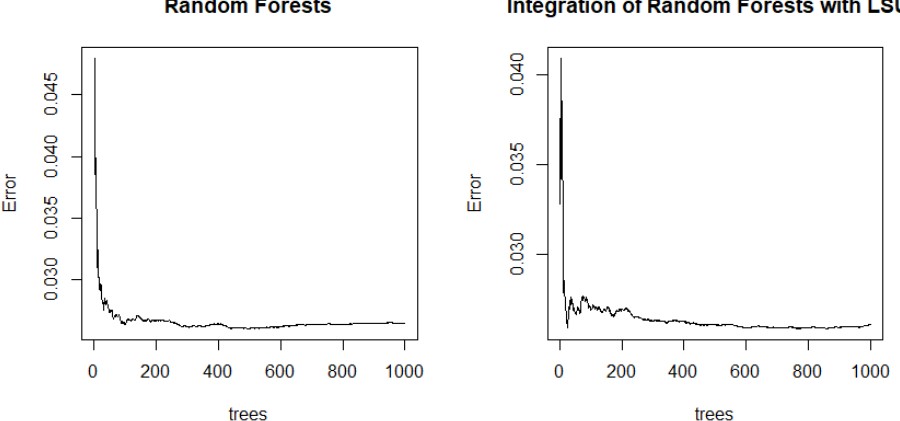

**Figure 7.** Optimization of the number of trees utilized in Random Forests (**left**) and the integration of Random Forests with LSUA (**right**).

Incorporating field measurements of carbon density as the dependent variable and using variables selected by LSR and LMSR as independent variables, RF achieved a coefficient of determination ($R^2$) of 0.7630, with plot mean and map mean estimates of 15.326 Mg/ha and 15.419 Mg/ha, respectively, and an RMSE of 8.741 Mg/ha. Integration of RF with LSUA improved performance, resulting in an $R^2$ of 0.7800, plot mean estimate and map mean of 15.207 Mg/ha and 15.136 Mg/ha, respectively, bringing them closer to the field measurement mean.

Both the RF model and the integration of RF with LSUA effectively depicted the spatial distribution of vegetation carbon density across the study area (Figure 6i,j). Both models tended to overestimate areas with 0 Mg/ha and underestimate regions with very high vegetation carbon density. The RF model combined with LSUA demonstrated improved stability and provided more detailed information on vegetation carbon in mixed pixels, requiring a greater number of trees for enhanced performance compared to the RF model without vegetation fraction from LSUA.

## 5. Discussion

Urban vegetation, comprising forests, shrubs, and grass, plays a vital role in mitigating atmospheric carbon concentration through processes such as photosynthesis. It contributes to air purification, noise reduction, and climate change impact mitigation [60–67]. The accurate mapping of urban vegetation carbon density is critical for governmental planning and residents' well-being. Our study presents a methodological framework for mapping vegetation carbon density in urban areas, utilizing a combination of field plot measurements, Landsat 8 imagery, and Pleiades 1A and 1B imagery data. While the integration of diverse data sources for mapping biomass or carbon density is not new in forested areas [68–70], our research addresses unique challenges in the urban environment. Shadows from buildings and mountains pose difficulties in accurately extracting vegetation information, compounded by the complex and fragmented urban landscape, leading to numerous mixed pixels that complicate the precise mapping of urban vegetation carbon density.

In this study, we employed LSUA to remove shadows induced by buildings and mountains from Landsat 8 images before spatial modeling. The method was evaluated using a random sample of 300 pixels across the study area, including 29 pixels located in mountain-induced shadow areas. While LSUA proved effective in mountainous regions, its performance was somewhat limited in urban areas due to the coarse spatial resolution of Landsat 8. After shadow removal, correlation coefficients between plot vegetation carbon density and shadow-removed Landsat 8 bands increased by 1.28% to 2.59% across different bands compared to the unprocessed Landsat 8 image. Our results align with findings from other studies employing LSUA for shadow removal and vegetation analysis [71,72]. Furthermore, our study validated LSUA by decomposing mixed pixels and extracting vegetation fractional information, achieving the highest correlation coefficient of 0.595 between the vegetation fractional image and plot carbon density when using four endmembers with automatic selection after shadow removal. This surpasses results obtained through manual selection, highlighting the effectiveness of LSUA in improving the accuracy of vegetation information extraction.

Spatial interpolation methods, including parametric (LSR, LMSR) and non-parametric (kNN, DT, RF) models, were utilized for mapping urban vegetation carbon density. The utilization of parametric and non-parametric models allows for a nuanced exploration of the complex relationships governing urban vegetation carbon density. While parametric models provide straightforward insights into linear relationships, non-parametric models, especially Decision Trees and Random Forests, offer a more flexible and adaptive approach, crucial for capturing the heterogeneity inherent in urban landscapes. The integration of diverse models aims to capitalize on their respective strengths, compensating for limitations in individual approaches. It is important to note that the selection of these models is not arbitrary but based on their suitability for addressing the specific challenges outlined in the study, emphasizing the need for a versatile and robust methodology in urban carbon

mapping. The ensuing comparative analysis of these models provides a comprehensive understanding of their performance, contributing valuable insights to the field of remote sensing and urban ecology.

The study employed $R^2$ and RMSE as performance metrics, revealing that Decision Trees consistently outperformed other models with the highest $R^2$ and the lowest RMSE, indicative of superior accuracy. The integration of LSUA enhanced prediction accuracies across models, although the improvements were not statistically significant, suggesting LSUA's role in refining models without a substantial boost in accuracy. Relative errors were compared among models, with Decision Trees and Random Forests consistently demonstrating lower errors, underscoring their reliability in capturing the complexities of the urban landscape. In the comparison between Decision Trees and Random Forests, Decision Trees, especially when integrated with LSUA, exhibited superior performance, addressing overfitting concerns and resulting in a slight improvement in correlation. Integration of these models with vegetation fractional images from LSUA improved mapping accuracy by 0.20% to 2.70%, depending on the model used. This enhancement, though relatively modest compared to some studies, signifies the potential for increased accuracy in mapping urban vegetation carbon density, especially in mixed environments with varying land features. The $R^2$ and RMSE results revealed that the DT model exhibited the best performance, demonstrating the highest $R^2$ and lowest RMSE among all models, whether integrated with vegetation fraction images from LSUA or not. LSR, regardless of LSUA integration, yielded extremely large and illogically negative estimates. Both LMSR and its integration with LSUA produced consistent estimates within the range of field plot values. kNN and kNN with LSUA tended to underestimate carbon density for large values and overestimate for small values. Relative RMSE values for urban vegetation carbon density estimates were 72.4% and 72.1% for LSR and LSR with LSUA, 61.4% and 60.3% for LMSR and LMSR with LSUA, 70.5% and 64.5% for kNN and kNN with LSUA, 58.3% and 57.7% for RF and RF with LSUA, and 46.35% and 45.87% for DT and DT with LSUA, respectively. The DT and DT with LSUA demonstrated the best performance, followed by RF and RF with LSUA, LMSR and LMSR with LSUA, and kNN and kNN with LSUA. LSUA integration improved prediction accuracies, albeit not significantly.

Compared to Yan et al. (2015) [73], the relative errors in this study were larger due to a higher coefficient of variation (108.87%) in urban vegetation carbon density resulting from a complex landscape, mixed pixels, and building-induced shadows. The dominance of built-up areas in Shenzhen city, as opposed to the forested areas in Yan et al.'s study, contributed to these differences. While kNN has shown effectiveness in mapping forest parameters in other studies [54,74–77], its application to estimate urban vegetation carbon density in this research resulted in lower accuracy compared to other methods. This discrepancy arises from the spectral distance-based approach of kNN, which encounters challenges in urban areas with mixed pixels exhibiting varied spectral reflectance due to diverse structures and compositions. The DT and DT with LSUA demonstrated strong predictive capabilities for urban vegetation carbon density, achieving high correlations of 0.8171 and 0.8205, respectively. The slight improvement in correlation with LSUA integration suggests that the DT model effectively mitigated overfitting issues. RF and RF with LSUA also exhibited high predictive capabilities, with RF integrated with LSUA identified as the more promising model for accurately mapping urban vegetation carbon density. This method holds great potential for rapid and accurate mapping with an acceptable level of error and cost-effectiveness.

In this study, although very high spatial resolution Pleiades 1A and 1B images were acquired, they were solely used for stratified random sampling and validating vegetation fraction images derived from LSUA. Despite Pleiades images providing superior visual interpretation capabilities compared to Landsat 8, their disadvantages include increased internal variability within homogeneous land cover polygons and limited spectral resolution with only four bands. The fine spatial resolution of Pleiades images also demands extensive storage and high-performance computation. Consequently, due to cost constraints,

Pleiades 1A and 1B images were not utilized for mapping vegetation carbon density in this study. The aim of this research was to establish a cost-efficient methodology for mapping urban vegetation carbon density in developing countries. The high cost of Pleiades images, amounting to $58,608 (USD) for Shenzhen city coverage, renders their use impractical for mapping vegetation carbon density in large Chinese cities or in the regional scale. Utilizing freely available Landsat 8 images emerges as the optimal option, significantly reducing research costs. Future studies should investigate the impact of spatial and spectral resolutions on the accuracy of estimating urban vegetation carbon density. Urban sprawl induces land use and land cover (LULC) conversion, influencing total carbon stock and carbon pools. Previous studies highlighted the proportionate increase in anthropogenic carbon stock and decline in soil and vegetation carbon stock in populated areas [78–81]. While this study concentrated on mapping urban vegetation carbon density, future research should address and assess the effects of urbanization on carbon pool dynamics, considering data availability.

The study establishes a methodological framework to enhance the precision of urban vegetation carbon density mapping through spatial modeling, spectral unmixing analysis, and de-shadowing techniques. The validated efficacy of this framework indicates improved accuracy. To advance the field, future studies should prioritize innovative methods for refining vegetation information extraction from mixed pixels and mitigating shadow impacts. A promising avenue for accuracy enhancement lies in exploring data fusion techniques, especially by incorporating optical imagery with LiDAR and RADAR data. Further research directions involve exploring additional machine learning models, extending the methodology to diverse urban environments, and addressing challenges like spatial resolution limitations. Insights into the effects of urbanization on carbon dynamics and anthropogenic activities significantly contribute to a broader understanding of urban carbon mapping.

## 6. Conclusions

The study focuses on accurately mapping urban above-ground vegetation carbon density, addressing challenges posed by complex urban landscapes, mixed pixels, and building-induced shadows. A novel methodological framework is introduced, combining linear spectral unmixing analysis (LSUA) for shadow removal, spatial modeling, and integration of diverse data sources, including Landsat 8, Pleiades 1A and 1B, DEM, and field measurements. The shadow removal algorithm effectively operates in mountainous areas but shows limitations in urban settings due to Landsat 8's coarse spatial resolution. LSUA improves correlation after shadow removal, and integration with spatial models enhances mapping accuracy, with Decision Trees exhibiting superior performance. While relative improvements are modest, the potential for increased accuracy in mapping urban vegetation carbon density is highlighted. Despite challenges, the DT model, especially integrated with LSUA, demonstrates the best performance. However, relative errors are larger compared to a similar study, attributing this to the complex urban landscape. Cost considerations favor the use of freely available Landsat 8 images over higher-cost Pleiades images. Future research should explore the impact of spatial and spectral resolutions, assess urbanization effects on carbon pools, and develop novel methods for vegetation information extraction and shadow mitigation. Overall, the study provides a cost-efficient methodology with potential for accurate urban vegetation carbon density mapping.

**Author Contributions:** Conceptualization, G.Q., G.W. and M.W.; methodology, G.Q.; software, G.Q.; validation, G.W., M.W. and J.Y.; formal analysis, G.Q.; investigation, G.W.; resources, G.Q.; data curation, G.Q.; writing—original draft preparation, G.Q.; writing—review and editing, G.W. and M.W.; visualization, G.Q. and G.W.; supervision, G.W.; project administration, G.W., G.Q. and M.W.; funding acquisition, G.W., G.Q. and M.W. All authors have read and agreed to the published version of the manuscript.

**Funding:** This study was supported by the research project "Shenzhen vegetation biomass and carbon modeling" awarded by Shenzhen Xianhu Botanic Garden, grant number #8851; Municipal Science and Technology Cooperation, grant number (2022) 163; Moutai College High-Level Talents Research Initiation Fund Project, grant number (2022) 134; Moutai College High-Level Talents Research Initiation Fund Project, grant number (2022) 049.

**Data Availability Statement:** All data will be available upon request.

**Acknowledgments:** Authors appreciate the assistance of Yifan Tan and Hua Sun on the field data collection and funding from the Shenzhen Xianhu Botanic Garden, Shenzhen, China, Department of Tourism of Management, Moutai Institute, China, and we appreciate all the anonymous reviewers of this article who provided valuable revision comments.

**Conflicts of Interest:** The authors declare no conflicts of interest. The funding sponsors had no role in the design of the study; in the collection, analyses, or interpretation of data; in the writing of the manuscript, and in the decision to publish the results.

## Appendix A

**Table A1.** Tree volume calculation equations.

| Tree Species | Volume Calculation Equation |
|---|---|
| Eucalypts | $V = 8.71419 \times 10^{-5}D^{1.94801}H^{0.74929}$ |
| Pinus elliottii | $V = 7.81515 \times 10^{-5}D^{1.79967}H^{0.98178}$ |
| Acacia rachii | $V = 7.32715 \times 10^{-5}D^{1.65483}H^{1.08069}$ |
| Chinese red pine | $V = 7.98524 \times 10^{-5}D^{1.74220}H^{1.01198}$ |
| Castanopsis fissa | $V = 6.29692 \times 10^{-5}D^{1.81296}H^{1.01545}$ |
| Broad-leaved | $V = 6.74286 \times 10^{-5}D^{1.87657}H^{0.92888}$ |
| Cunninghamin lanceolata | $V = 6.97483 \times 10^{-5}D^{1.81583}H^{0.99610}$ |
| Hard latissimus | $V = 6.01228 \times 10^{-5}D^{1.87550}H^{0.98496}$ |

Note: V—tree volume, D—diameter at breast height (1.3 m), H—height of tree.

## Appendix B

**Table A2.** Biomass and volume relationship parameter values table.

| Forest Types | a (Mg/m$^3$) | b (Mg) | N | R$^2$ |
|---|---|---|---|---|
| Picea asperata Mast/Abies alba | 0.5519 | 48.861 | 24 | 0.78 |
| Bethula | 1.0687 | 10.237 | 9 | 0.70 |
| Casuarinaequisetifolia | 0.7441 | 3.2377 | 10 | 0.95 |
| Cunninghamialanceaolata | 0.4652 | 19.141 | 90 | 0.94 |
| Cedarwood | 0.8893 | 7.3965 | 19 | 0.87 |
| Cupressusfunebris | 1.1453 | 8.5473 | 12 | 0.98 |
| Quercus subg Quercus sect | 0.8873 | 4.5539 | 20 | 0.8 |
| Eucalyptus robusta smith | 0.6096 | 33.806 | 34 | 0.82 |
| Larixprinchipis-rupprechtii | 0.9292 | 6.494 | 24 | 0.83 |
| Subtropical evergreen broad-leaved forest | 0.8136 | 18.466 | 10 | 0.99 |
| Theropencedrymion | 0.9788 | 5.3764 | 35 | 0.93 |
| Broadleaf mixed plantations | 0.5856 | 18.744 | 9 | 0.91 |
| Pinus armandi | 0.5723 | 16.489 | 22 | 0.93 |
| Pinusmassoniana | 0.5034 | 20.547 | 52 | 0.87 |
| Sylvestris/Pinus | 1.112 | 2.6951 | 15 | 0.85 |

**Table A2.** *Cont.*

| Forest Types | a (Mg/m$^3$) | b (Mg) | N | R$^2$ |
|---|---|---|---|---|
| Pinustabuliformis | 0.869 | 9.1212 | 112 | 0.91 |
| Others Conifer | 0.5292 | 25.087 | 19 | 0.86 |
| Aspen | 0.4969 | 26.973 | 13 | 0.92 |
| Tsugachinensis/Criptomeriafortunei | 0.3491 | 39.816 | 30 | 0.79 |
| Tropical forests | 0.7975 | 0.4204 | 18 | 0.87 |

**Appendix C**

**Table A3.** Carbon ratio table for different tree species in China.

| Trees Species | Ratio | Tree Species | Ratio |
|---|---|---|---|
| Picea asperata Mas | 0.4994 | Schima | 0.5115 |
| Tsuga chinensis | 0.5022 | Others broad-leaved hard wood | 0.4901 |
| Larix gmelinii | 0.5137 | Aspen | 0.4502 |
| Pinus koraiensis Sieb | 0.5113 | Eucalyptus | 0.4748 |
| Pinus thunbergii Parl | 0.5146 | Acacia rachii | 0.4666 |
| Pinus tabulaeformis | 0.5184 | Others broad-leaved soft wood | 0.4502 |
| Pinus armandii Franch | 0.5177 | Broadleaf mixed trees | 0.4796 |
| Pinus massoniana Lamb | 0.5271 | Economic trees | 0.4700 |
| Pinus elliotii | 0.5311 | Cupressus funebris Endl | 0.5088 |
| Others Pinus | 0.4963 | Coniferous mixed forest | 0.5168 |
| Cunninghamia lanceolate | 0.5127 | * Bush | 0.4672 |
| Conifer-broadleaf forest | 0.4893 | * Herbal | 0.3270 |

Note: * Bush is a joint name of all kinds of different shrub species, * Herbal is a joint name of all kinds of different grass species.

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
