# Peer review of "Enhancing Urban Above-Ground Vegetation Carbon Density Mapping: An Integrated Approach Incorporating De-Shadowing, Spectral Unmixing, and Machine Learning"

_forests, doi:10.3390/f15030480_

Round 1

Reviewer 1 Report

Comments and Suggestions for Authors

The manuscript "Enhancing Urban Above-ground Vegetation Carbon Density Mapping: An Integrated Approach Incorporating De-shadowing, Spectral Unmixing, and Machine Learning" presents a comprehensive study aiming to improve the accuracy of mapping urban vegetation carbon density. The authors introduce a novel methodological framework that combines linear spectral unmixing analysis for shadow removal and vegetation information extraction, with a comparative analysis of parametric and non-parametric models including linear regression, k-Nearest Neighbors, Decision Trees, and Random Forests. Applied to Shenzhen, China, this study integrates various data sources, demonstrating the effectiveness of shadow removal in mountainous areas and marginal improvements in carbon density mapping with enhanced models. The integration of machine learning algorithms, particularly Decision Trees with linear spectral unmixing analysis, shows promising results for advancing urban vegetation carbon density mapping's accuracy.

General Comments:

  1. Concept and Novelty: The concept of integrating shadow removal, spectral unmixing, and diverse machine learning models for enhancing urban vegetation carbon density mapping is innovative and timely, addressing the challenges posed by urban landscapes for carbon density estimation.

  2. Methodology: The methodology is robust, combining various techniques and datasets to improve mapping accuracy. However, the manuscript could benefit from a more detailed discussion of the selection criteria for models and their comparative analysis.

  3. Data and Reproducibility: The use of openly available datasets alongside detailed methodological descriptions enhances the study's reproducibility. Nevertheless, additional information on data preprocessing steps could further aid in replicating the results.

Specific Comments:

  1. Shadow Removal Effectiveness: The effectiveness of the shadow removal algorithm, particularly in mountainous areas, is a key contribution. It would be beneficial to quantify the improvements in accuracy more explicitly, perhaps through specific metrics or comparative analyses.

  2. Model Performance Comparison: While the manuscript discusses the comparative performance of different models, it lacks detailed statistical analysis or error metrics that would clearly demonstrate the superiority of one model over others. Including a more comprehensive statistical comparison would strengthen the conclusions drawn.

  3. Integration of Techniques: The integration of spectral unmixing and machine learning models is highlighted as a novel approach. Clarifying how these methods complement each other and the specific improvements attributed to this integration would provide deeper insights into the study's innovation.

  4. Future Directions: The discussion on potential areas for future research, such as the exploration of additional machine learning models or the application of the methodology to different urban environments, would enrich the manuscript. Insights into limitations and future directions could be more prominently discussed to guide subsequent research efforts.

Conclusion:

This study makes a significant contribution to the field of urban vegetation carbon density mapping by introducing an integrated approach that enhances accuracy through the innovative use of shadow removal, spectral unmixing, and machine learning. While the manuscript presents promising results, elaboration on model selection, detailed statistical comparisons, and further insights into data preprocessing could strengthen its impact. Future research directions could also be more explicitly outlined to inspire continued advancements in this important area of study.

Comments on the Quality of English Language

The manuscript needs further proofreading.

Reviewer 2 Report

Comments and Suggestions for Authors    

In general, the manuscript is well-written and has a clear storyline with highlighted novelty. Below are my minor comments to further improve the quality of the study:

L16: Is it "mixed pixels" or "mixed spectral signatures"? L16: Does every city have high buildings and mountains? Please rewrite this sentence. L48: How about factors related to remote sensing (RS) data? This needs to be added to this section. L57: Why only Landsat? What about Sentinel-2 and other commercially available satellites like PlanetScope, SPOT…?

L117: You started directly with "(situated)," and I think it would be better to say something like "(the study area or the test site is situated or located…)."

L167: I'm not sure why the sample plots are designed as 2 by 2 meters, while the Landsat data has a 30m spatial resolution?

L175-179: To me, this is repeated and not needed again.

L212: Why were radiometric and geometric correction used when the Landsat 8 data can be obtained fully processed from the USGS service? Which Landsat level has been used?

Figure 3: It's very hard to read the legend on this figure and understand. Please correct.

L424-426: not needed.

Decision: This section needs a bit of work to include more studies to compare and justify your main findings. In its current form, it mainly explains the results without comparing them with existing literature.
